# Variation in outcome of invasive mechanical ventilation between different countries for patients with severe COVID-19: A systematic review and meta-analysis

**Hany Hasan Elsayed**[1]*, **Aly Sherif Hassaballa**[2], **Taha Aly Ahmed**[2], **Mohammed Gumaa**[3], **Hazem Youssef Sharkawy**[2], **Assem Adel Moharram**[4]

**1** Thoracic Surgery Department (ARDS Taskforce), Ain Shams University, Cairo, Egypt, **2** Cardiothoracic Surgery Department, Ain Shams University, Cairo, Egypt, **3** TRUST Research Center, Cairo, Egypt, **4** Anaethesia, Pain and Intensive Care Department, Ain Shams University, Cairo, Egypt

* hanyhassan77@hotmail.com

**Data Availability Statement:** All relevant data are within the paper and its Supporting Information files.

## Abstract

### Background

COVID 19 is the most recent cause of Adult respiratory distress syndrome ARDS. Invasive mechanical ventilation IMV can support gas exchange in patients failing non-invasive ventilation, but its reported outcome is highly variable between countries. We conducted a systematic review and meta-analysis on IMV for COVID-associated ARDS to study its outcome among different countries.

### Methods

CENTRAL, MEDLINE/PubMed, Cochrane Library, and Scopus were systematically searched up to August 8, 2020. Studies reporting five or more patients with end point outcome for severe COVID 19 infection treated with IMV were included. The main outcome assessed was mortality. Baseline, procedural, outcome, and validity data were systematically appraised and pooled with random-effect methods. Subgroup analysis for different countries was performed. Meta-regression for the effect of study timing and patient age and were tested. Publication bias was examined. This trial was registered with PROSPERO under registration number CRD42020190365.

### Findings

Our electronic search retrieved 4770 citations, 103 of which were selected for full-text review. Twenty-one studies with a combined population of 37359 patients with COVID-19 fulfilled the inclusion criteria. From this population, 5800 patients were treated by invasive mechanical ventilation. Out of those, 3301 patients reached an endpoint of ICU discharge or death after invasive mechanical ventilation while the rest were still in the ICU. Mortality from IMV was highly variable among the included studies ranging between 21% and 100%. Random-effect pooled estimates suggested an overall in-hospital mortality risk ratio of 0.70

**Funding:** The authors received no specific funding for this work.

**Competing interests:** The authors have declared that no competing interests exist.

**Abbreviations:** ARDS, Adult respiratory distress syndrome; IMV, Invasive mechanical ventilation; ICU, Intensive care unit; MERS, Middle east respiratory syndrome.

(95% confidence interval 0.608 to 0.797; I2 = 98%). Subgroup analysis according to country of origin showed homogeneity in the 8 Chinese studies with high pooled mortality risk ratio of 0.97 (I2 = 24%, p = 0.23) (95% CI = 0.94–1.00), similar to Italy with a low pooled mortality risk ratio of 0.26 (95% CI 0.08–0.43) with homogeneity (p = 0.86) while the later larger studies coming from the USA showed pooled estimate mortality risk ratio of 0.60 (95% CI 0.43–0.76) with persistent heterogeneity (I2 = 98%, p<0.001). Meta-regression showed that outcome from IMV improved with time (p<0.001). Age had no statistically significant effect on mortality (p = 0.102). Publication bias was excluded by visualizing the funnel plot of standard error, Egger's test with p = 0.714 and Begg&Mazumdar test with p = 0.334.

## Interpretation

The study included the largest number of patients with outcome findings of IMV in this current pandemic. Our findings showed that the use of IMV for selected COVID 19 patients with severe ARDS carries a high mortality, but outcome has improved over the last few months and in more recent studies. The results should encourage physicians to use this facility when indicated for severely ill COVID-19 patients.

## Introduction

Coronavirus disease 2019 (COVID-19) is a viral respiratory tract infection caused by a coronavirus which was first documented in Wuhan, China, in December 2019 [1]

After then, this outbreak spread globally and has been considered as a pandemic and an international public health emergency by the WHO on March 11, 2020. As of 1st of May 2021, a cumulative total of around 151 million confirmed cases of coronavirus disease 2019 (COVID-19) were reported with total 3.17 million deaths in 203 countries and territories worldwide [2]. Currently, there is no proven effective medication discovered for the COVID-19 infection.

The FDA has approved emergency use permission for the Pfizer/BioNTech and Moderna COVID-19 vaccines. These vaccines can protect receivers from a SARS-CoV- 2 infection by development of antibodies and afford immunity against a SARS-CoV-2 infection. Both vaccines can develop various adverse effects, but reported to be less frequent in the Pfizer/BioNTech vaccine, however, the Moderna vaccine is easier to transfer and store since it is less temperature sensitive [3].

Although most patients with COVID-19 infection have only mild or uncomplicated course, around 10–20% will develop a severe disease that necessitates hospitalization and oxygen therapy or even ICU admission and progression to acute respiratory distress syndrome (ARDS). The prevalence of ARDS caused by COVID-19 is around 8.2% who will require mechanical ventilation and prone positioning [4].

Invasive mechanical ventilation (IMV) is life-saving in patients with severe respiratory failure not responding to other less invasive modalities. While the majority of COVID-19 patients can be successfully managed with oxygen therapy and/or non-invasive mechanical ventilation, patients with the most severe respiratory failure demand insertion of an endotracheal tube. Although an endotracheal tube enables control over an un-stable airway and facilitates precise regulation of oxygen, volume and pressure but inevitably, the endotracheal tube brings in its rouse a list of complications aggravated by the morbidity of the patient's other system failures.

Each extra day of invasive mechanical ventilation IMV exposes patients to more complications and increases mortality [5].

Lim et al. reported that almost half of patients with COVID-19 receiving IMV died but there was high variability between studies of the method of reporting case fatality. A higher mortality was noticed in older patients and in the early pandemic period which they attributed to possible limited ICU resources [6].

In view of the current growing pandemic and the fact that there is high variability in published data for the outcome of invasive mechanical ventilation IMV to support COVID-19 patients, we aimed to describe the mortality outcomes of a large group of COVID-19 patients who, due to the severity of their disease, required application of invasive mechanical ventilation. All available studies were utilised by performing a systematic review and meta-analysis.

## Methods

### Search strategy and selection criteria

This systematic review complies with the Meta-analysis Of Observational Studies in Epidemiology (MOOSE) and Preferred Reporting Items for Systematic Reviews and Meta-Analyses (PRISMA) guidelines. We electronically ran a search on CENTRAL, MEDLINE/PubMed, Cochrane Library, and Scopus. On Pubmed the word search used was (COVID OR SARS COV2 OR pandemic) AND (ARDS) OR (acute respiratory distress syndrome) (Intensive care unit) OR (ICU) OR (Intensive therapy unit) OR (ITU) OR (acute lung injury) OR (respiratory failure) OR (respiratory insufficiency) OR (mechanical ventilation) OR (invasive ventilation)).

We searched trial registries, included the grey literature, and used studies accepted and ahead of print. We did our search up to 8th of August 2020 without language restrictions. We used both subject headings and text word terms to search for articles about mechanical ventilation with ARDS in COVID-19 patients. Inclusion criteria were (all criteria should be concomitantly met for study inclusion): a) study reporting on 5 or more patients with final outcomes; b) with confirmed COVID 19 infection; c) receiving invasive mechanical ventilation. Exclusion criteria were (one criterion was sufficient for study exclusion): a) inclusion of <5 patients with COVID-19 infection treated with IMV (thereby, any study reporting on fewer than 5 patients or case reports treated with IMV were excluded); b) duplicate publication (in which case only the most recent report from the same study group was included in the systematic review). Use of a sample size cut-off was chosen to limit the risk of imprecision and publication bias c) studies with insufficient data about the outcome endpoints (mortality and ICU discharge). AH, TA and HY independently reviewed the titles and abstracts of all citations. Then, they independently reviewed the full text of both definite and potentially eligible studies for inclusion. Disagreements were reviewed by a fourth reviewer HE, who had a deciding vote. The study protocol link is at www.crd.yorl.ac.uk/PROSPERO under registration number CRD42020190365

### Patient and public involvement

No patients were involved in the study.

### Data analysis

A single arm meta-analysis was conducted to examine the mortality incidence in invasive mechanical ventilation treatment for COVID 19. Data were summarized using the risk ratio (95% confidence interval (CI)). The data were pooled using DerSimonian-Laird random effects model [7]. P value of 0.05 or less was statistically significant. Cochran Q and I2 were

used to assess heterogeneity between studies. The degree of heterogeneity was categorized as either low (I2 < 25%), moderate (I2 = 25%–75%), or high (I2 > 75%) [8]. A P value of ≤ 0.05 indicated significant heterogeneity. A subgroup meta-analysis according to the study's country of origin was conducted to investigate the high heterogeneity detected. Two additional sensitivity analysis were conducted; one meta-analysis included only the six studies that reported complete outcome endpoints for all the patients who received IMV. The second meta-analysis included only the studies that enrolled more than 100 patients. The data used in the meta-analysis in each study were the number of mortality events and the number of closed cases (either ICU discharged or dead). The study of the outcome in relation to the time of each study was performed by calculating a median number representing the central timing of the study. Timing was calculated over 180 days (from 1–180) starting at the beginning of December 2020 to the end of May 2020 (6 x 30 days) and each study was given a range of days starting from the first to the last day as recorded in the study duration. A median number was calculated for the time range of each study (available for 20 out of 21 studies). This was done to avoid time overlap and duration bias between different studies. Publication bias was examined by visual inspection of the funnel plot and tested by Egger's test and Begg and Mazumdar test. A P value of ≤ 0.05 indicated the existence of publication bias. All analyses were performed using Open Meta Analyst software Windows 10 version.

### Role of funding source

There was no funding source for this study. The corresponding author of this study had full access to all the study data and had final responsibility for the decision to submit the manuscript for publication.

## Results

Our electronic search retrieved 4770 citations, 103 of which were selected for full-text review (**Fig 1**). Twenty-one studies [9–29] with a combined population of 37359 patients fulfilled the inclusion criteria. From this population, 5800 patients were treated by invasive mechanical ventilation. Out of those, 3301 patients reached an endpoint of ICU discharge or death after invasive mechanical ventilation while the rest were still in the ICU (regardless of mechanical ventilation state). All studies are summarized in **Table 1**.

Mortality from IMV was highly variable among the included studies ranging between 21% and 100%. Random-effect pooled estimates suggested an overall in-hospital mortality risk ratio of 0.70 (95% confidence interval 0.608 to 0.797; I2 = 98%) (**Fig 2**). Most of the preliminary studies were from China (eight studies with 203 patients with endpoints). Larger studies then followed from the USA, Italy, Denmark, UK, Canada, Japan and France (thirteen studies with 3098 patients with endpoints). Only six studies (29%) [11, 16, 18–21] reported complete outcome endpoints for all patients who received IMV while the rest of studies had patients who did not reach an endpoint.

To investigate the overall inter-study heterogeneity, a subgroup analysis was performed according to the country of origin of each study (**Fig 3**) This showed homogeneity in the 8 Chinese studies with high pooled mortality risk ratio of 0.97 (I2 = 24%, p = 0.23) (95% CI = 0.94–1.00), similar to Italy with a low pooled mortality risk ratio of 0.26 (95% CI 0.08–0.43) with homogeneity (p = 0.86) while the later larger studies coming from the USA showed pooled estimate mortality risk ratio of 0.60 (95% CI 0.43–0.76) with persistent heterogeneity (I2 = 98%, p<0.001).

In the studies that reported complete outcome endpoints for all the enrolled patients, mortality risk from IMV was 0.943 (95% CI: 0.889 to 0.997) with moderate non-significant

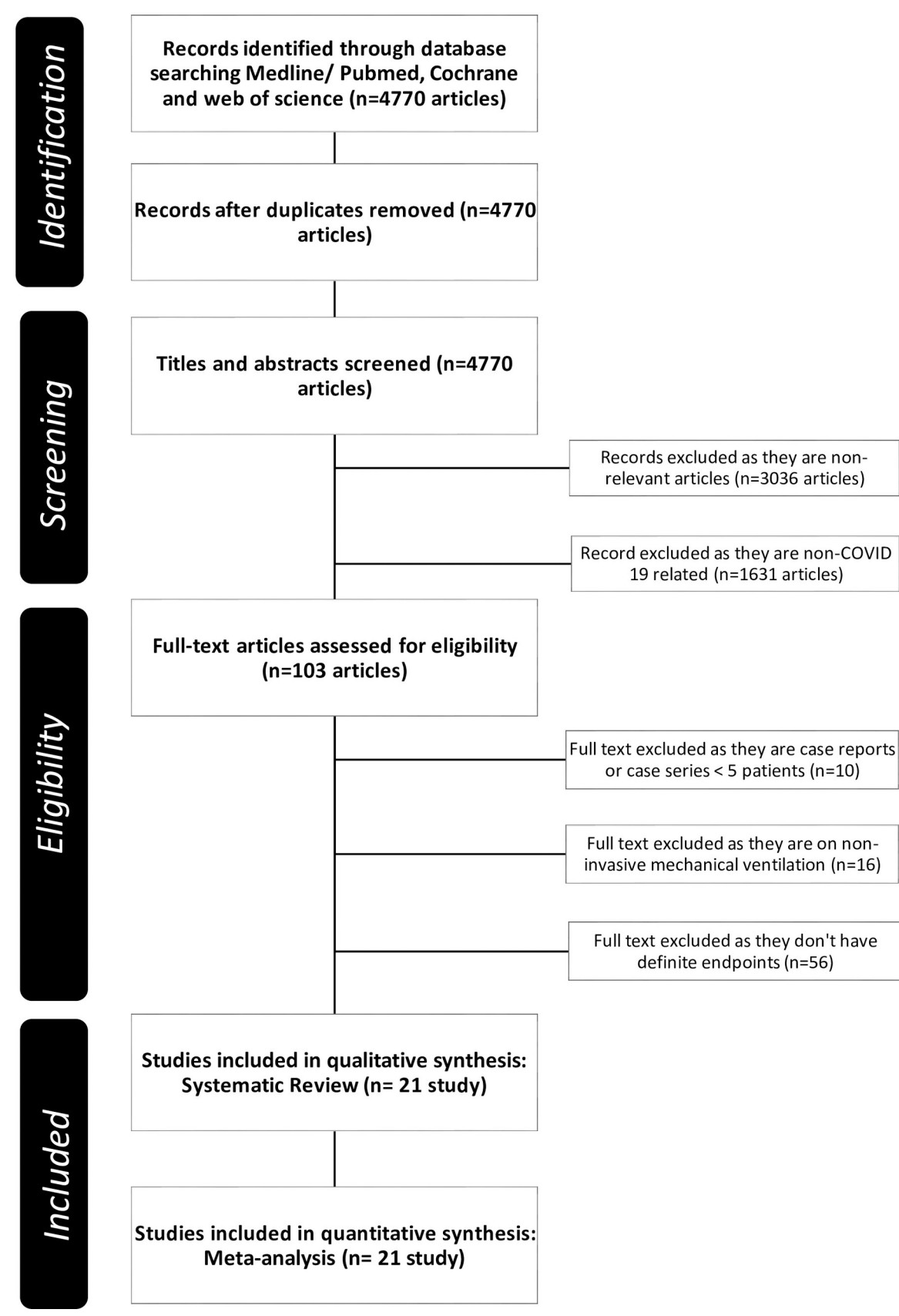

**Fig 1. PRISMA flow diagram of study selection.**

**Table 1. All studies included in the meta-analysis.**

| Author | Name | Country | Duration | Age | No. of patients | No. of ARDS | No. of IMV | Cure | Death |
|---|---|---|---|---|---|---|---|---|---|
| Wang et al. [9] | Tracheal intubation in patients with severe and critical COVID-19: analysis of 18 cases | China | From February 12th to February 28th | 70.39±8.02 | 18 | N/A | 18 | 0 | 5 |
| Pedersen et al. [10] | Initial experiences from patients with COVID-19 on ventilatory support in Denmark. | Denmark | From 11 March 2020 to 01 April 2020 | 69.5 years (range: 56–84 years) | 16 | 16 | 16 | 4 | 7 |
| Chen et al. [11] | Clinical characteristics of 113 deceased patients with coronavirus disease 2019: retrospective study | China | From 13 January to 12 February 2020 | 62.0 (44.0–70.0) | 274 | 196 | 17 | 0 | 17 |
| Yu et al. [12] | Patients with COVID-19 in 19 ICUs in Wuhan, China: a cross-sectional study | China | February 26 to 27, 2020 | 64 (57–70 | 226 | 161 | 121 | 0 | 79 |
| Richardson et al. [13] | Presenting Characteristics, Comorbidities, and Outcomes Among 5700 Patients Hospitalized With COVID-19 in the New York City Area | USA | From March 1, 2020 to April 4, 2020 | 63 years IQR 52–75; range | 5700 | N/A | 1151 | 38 | 282 |
| Grasseli et al. [14] | Baseline Characteristics and Outcomes of 1591 Patients Infected With SARS-CoV-2 Admitted to ICUs of the Lombardy Region, Italy | Italy | From February 20 and March 25, 2020 | 63 (56–70) years | 1591 | N/A | 1150 | 256 | 405 |
| Zangrillo et al. [15] | Characteristics, treatment, outcomes and cause of death of invasively ventilated patients with COVID-19 ARDS in Milan, Italy. | Italy | From 20 February to 2 April 2020 | 61years (interquartile range [IQR], 54–69. | 73 | 73 | 73 | 17 | 23 |
| Bhatraju et al. [16] | Covid-19 in Critically Ill Patients in the Seattle Region—Case Series | USA | From February 24 to March 9, 2020 | Mean 64±18 | 24 | 18 | 18 | 6 | 12 |
| Kato et al. [17] | Clinical course of 2019 novel coronavirus disease (COVID-19) in individuals present during the outbreak on the Diamond Princess cruise ship | Japan | From March 11 to March 19, 2020 | 76 | 70 | N/A | 14 | 7 | 2 |
| Wu et al. [18] | Risk Factors Associated with Acute Respiratory Distress Syndrome and Death in Patients with Coronavirus Disease 2019 Pneumonia in Wuhan, China | China | From December 25, 2019, to February 13, 2020 | 51 (43–60) | 210 | 84 | 5 | 0 | 5 |
| Yang et al. [19] | Clinical course and outcomes of critically ill patients with SARS-CoV-2 pneumonia in Wuhan, China: a single-centered, retrospective, observational study. | China | From late December 2019 to Jan 26, 2020 | 59.7 (13.3) | 52 | 52 | 22 | 3 | 19 |
| Ruan et al. [20] | Correction to: Clinical predictors of mortality due to COVID-19 based on an analysis of data of 150 patients from Wuhan, China | China | N/A | Survivors: 67 (15–81) Non-Survivors: 50 (44–81) | 150 | 62 | 25 | 0 | 25 |
| Zhou et al. [21] | Clinical course and risk factors for mortality of adult inpatients with COVID-19 in Wuhan, China: a retrospective cohort study. | China | From Dec 29, 2019 to Jan 31, 2020 | 56 (46–67) | female 72 (38%) Male 119 (62%) | 191 | 32 | 1 | 31 |
| Cummings et al. [22] | Epidemiology, Clinical Course, and Outcomes of Critically Ill Adults With COVID-19 in New York City: A Prospective Cohort Study | USA | From 2 March to 28 April, 2020. | 62 years (IQR 51–72) | 1150 | 257 | 203 | 58 | 84 |
| Xu et al. [23] | Risk factors for adverse clinical outcomes with COVID-19 in China: a multicenter, retrospective, observational study. | China | From January 10, 2020 and March 13 | 46.1 years (SD 15.2) (range from 2 months to 86 years old) | 382 males, 321 (46%) females | 703 | 20 | 4 | 14 |
| Argenziano et al. [24] | Characterization and clinical course of 1000 patients with coronavirus disease 2019 in New York: retrospective case series | USA | From 11 March to 6 April 2020 | 63.0 IQR (50.0–75.0) | Male 596; 59.6% Female 404; 40.4 | 1000 | 233 | 36 | 111 |

(*Continued*)

**Table 1.** (Continued)

| Author | Name | Country | Duration | Age | No. of patients | No. of ARDS | No. of IMV | Cure | Death |
|--------|------|---------|----------|-----|-----------------|-------------|------------|------|-------|
| *Mitra et al.* [25] | Baseline characteristics and outcomes of patients with COVID-19 admitted to intensive care units in Vancouver, Canada: a case series. | Canada | From Feb. 21 to Apr. 14, 2020 | 69 [IQR] 60–75 years | 38 (32.5%) were female | 117 | 74 | 34 | 15 |
| *Auld et a l (26)* | ICU and Ventilator Mortality Among Critically Ill Adults with Coronavirus Disease | USA | From March 6, 2020, to April 17, 2020 | 64 (54–73) | 98 (45.2) FEMALE | 217 | 165 | 88 | 59 |
| *Hur et al.* [27] | Factors Associated with Intubation and Prolonged Intubation in Hospitalized Patients With COVID-19 | USA | From 1 march to 8 April,2020. | 59 years (interquartile range, 47–69) | 486 | N/A | 138 | 78 | 21 |
| *Petrilli et al.* [28] | "Factors associated with hospital admission and critical illness among 5279 people with coronavirus disease 2019 in New York City: prospective cohort study | USA | From 1 March to 5 May,2020. | 54 years (interquartile range 38–66 years) | 5279 | N/A | 647 | 170 | 391 |
| *Docherty et al.* [29] | Features of 20 133 UK Patients in Hospital With covid-19 Using the ISARIC WHO Clinical Characterisation Protocol: Prospective Observational Cohort Study | United Kingdom | From 6 February to 3 May 2020 | 73 years (interquartile range 58–82) | 20 133 | N/A | 1658 | 276 | 618 |

heterogeneity between the studies (I2 = 48.06%; P = 0.087) (**Fig 4**). While in studies that included more than 100 patients, lower mortality risk was found 0.665 (95% CI: 0.570 to 0.759), however, the heterogeneity between the studies was still significantly high (I2 = 96.77%; P<0.001) (**Fig 5**).

To investigate the effect of the study time on the outcome, a meta-regression analysis of the median day of the study duration showed that mortality was lower as the study time was more recent (p<0.001) (**Fig 6**). Analysis of the effect of age on mortality showed no statistical significance (p = 0.102). Publication bias was excluded by visualizing the funnel plot of standard error (**Fig 7**), Egger's test with p = 0.714 and Begg&Mazumdar test with p = 0.334.

## Discussion

The mortality of IMV related for COVID-19 patients reported from different studies may vary according to the denominator used. For example, in the study by Richardson and colleagues

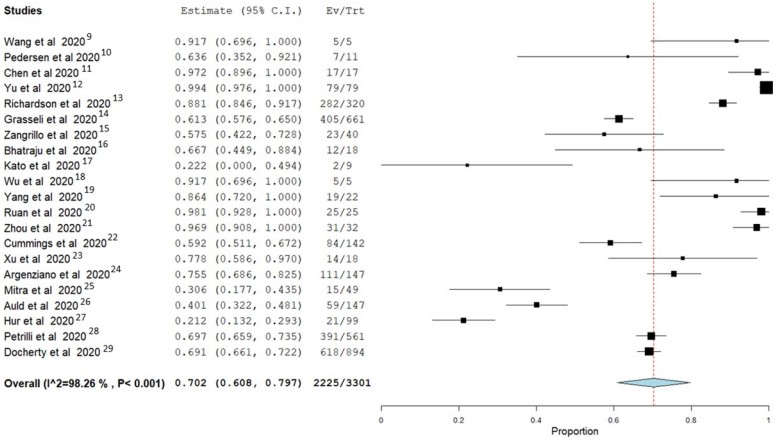

**Fig 2. Forest plot of pooled analysis of mortality by random effect model in all studies of IMV with COVID 19 patients.**

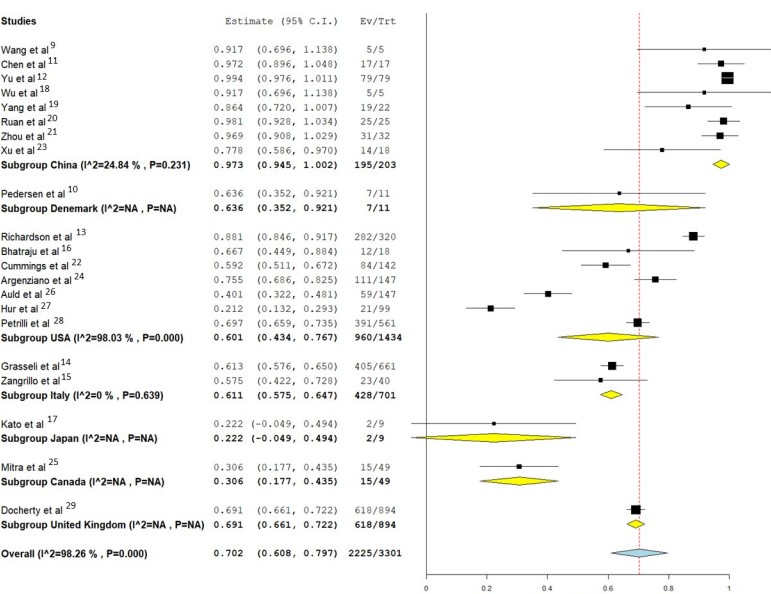

**Fig 3. Forest plot of pooled analysis of mortality by random effect model in all studies of IMV with COVID 19 patients with subgroup division of country of origin.**

from New-York, the denominators excluded patients who are still mechanically ventilated in ICU [13], but Graselli and his colleagues [14] from Italy, included those in the ICU in the denominator and the abstract for the data by Richardson and his colleagues has since been corrected to report the percentage of patients dead, alive, and still in the ICU to try to avoid this misinterpretation.

Preliminary reports from China regarding IMV for COVID-19 patients were obviously dismal as shown by our subgroup analysis with a median mortality of 97% (95% CI 94–100%). All the following studies apart from Richardson et al. [13] showed acceptable mortality rates (21%-69%) and this should change the perception for this crucial intervention. The misconception that all patients on IMV will die is the rule for all COVID-19 patients does not do good for anyone. Our headlines extolling that IMV mortality rate for COVID-19 are in the range of 90–100% makes the medical team wonder if it's worth risking their lives while dealing with such a futile intervention. The concern of families seeing their beloved ones ventilated after being infected will turn into nothing but terror. Additionally, countries with limited resources and ICU beds might not even bother procuring their ventilators in the era of an accelerating pandemic.

The outcome of IMV for patients with COVID-19 was not much worse than the previous respiratory virus pandemics. In a pooled mortality calculation from 3 studies [30–32] the

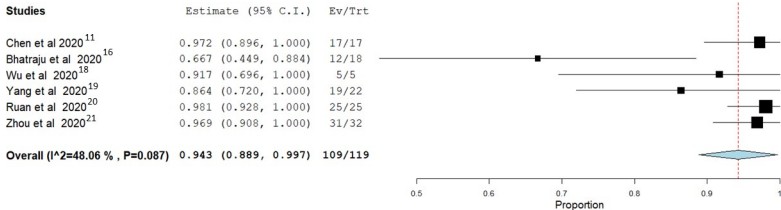

**Fig 4. Meta-analysis for the six studies that reported complete outcome endpoints for all the patients who received IMV.**

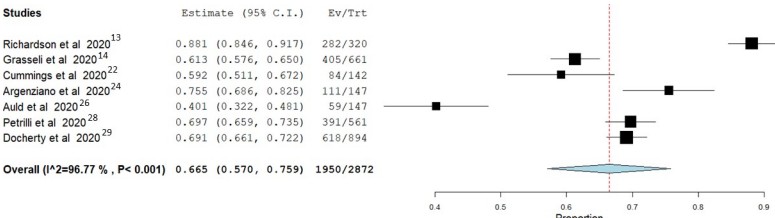

**Fig 5. Meta-analysis for the studies that enrolled more than 100 patients.**

outcome of invasive mechanical ventilation for treating patients with Middle East Respiratory syndrome MERS was a mortality of 77% (647/840).

Nevertheless, the mortality for patients with COVID-19 requiring IMV is worse than most common medical conditions requiring ICU admission. The reported ICU mortality from sepsis is 40% [33], severe COPD 30% [34] and pancreatitis 25% [35], all lower than COVID-19 patients requiring IMV in our study.

Timing of placing patients with severe COVID-19 on mechanical ventilation will vary. Mortality may have been lower if patients were placed on a ventilator earlier in their disease course. On the other hand, the denominator may be smaller where patients with respiratory failure were not offered mechanical ventilation in severe COVID-19. This may be due to family wishes, generous use of noninvasive ventilation or the scarcity of ventilator beds in a hastening pandemic. We have never been able to agree on universal triggers for ventilatory support, even with known diseases that are much better understood than COVID-19.

Mortality from COVID-19 has been reported to be age-dependent, and variations in population age or the age of admitted patients are likely to have a significant influence on mortality. Similar arguments may apply for comorbidities [36]. As we have only summary statistics, with variable reporting, we were unable to explore these factors in detail, though meta regression by the crude measure of average age was not significantly associated with reported mortality in our analysis. Reporting of such data in future cohort studies and trials would be beneficial.

The current indication to place a patient with severe COVID-19 on invasive mechanical ventilation is not a clear-cut one and neither are the outcomes. Most published reports (15/21 = 71%) in our analysis did not include full endpoints for all patients receiving IMV as they were still receiving IMV or still in the ICU so the assessment of the final outcome of IMV for these centres is not possible. We may not fully understand how or why these outcome data from each country look different but as our understanding of the COVID-19 improves over months, this may improve the outcome as our meta regression has shown.

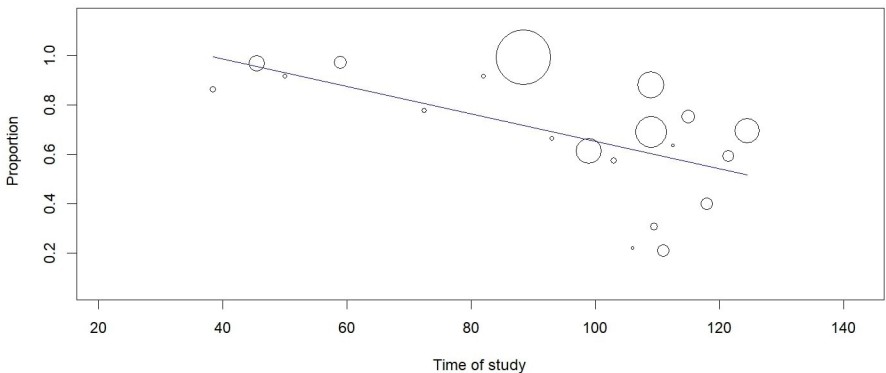

**Fig 6. Meta regression for the effect of time on mortality outcome in IMV for COVID-19.**

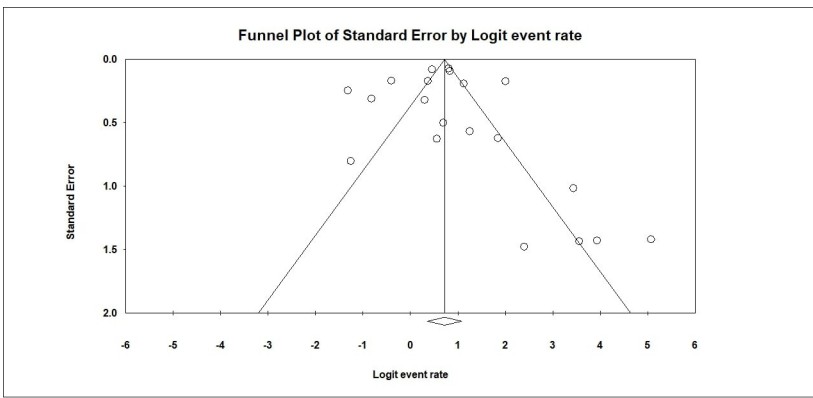

**Fig 7. Funnel plot of standard error.**

So, while we scrutinize these published reports and studies and try to extract what is universal to the medical community and can be applied to our understanding and care of patients locally, we need to recognize and report on the enormous drivers of differences and be vigilant in our presentation of data to minimise confusion in interpretation.

The variability of findings has always existed in studies of IMV for critically ill patients and COVID-19 is not an exception, merely an amplifier of these differences.

## Limitations

In our interpretation we excluded all patients still receiving care in the ICU to avoid the use estimated mortality with all its drawbacks and included only patients discharged from the ICU in the denominator. Obviously, this carries the risk of not including patients deteriorating outside the ICU after discharge or needing readmission after ICU discharge. A fraction of patients who survive ICU may die prior to hospital-discharge and the survival rate we reported will discreetly over-estimate survival to hospital-discharge. To try and put this into context, the long-running ICNARC case-mix registry reports a 5.7% in-hospital mortality rate for all patients after their discharge from ICU [37]. Whether this finding is replicated after ICU discharge in patients with COVID 19 is worthy of future research, together with the long-term outcomes of these patients. Bias from mortality results can be due different "inclusion/exclusion" criteria for mechanical ventilation across institutions resulting in different populations being compared and varying availability of resources including mechanical ventilators affecting outcomes in undetermined ways. Several studies in our analysis were excluded as they did not specifically report ICU outcome data; rather they included outcome data for the entire inpatient COVID-19 population, did not specify invasive mechanical ventilation patients or outcome data were not yet available on publication. It is possible that the ICU outcomes in these studies may have differed from the studies we were able to include in this analysis.

**In conclusion,** the study included the largest number of patients with outcome findings of IMV in this current pandemic. Our findings showed that the use of IMV for selected COVID 19 patients with severe ARDS carries a high mortality, but outcome has improved over the last few months and in more recent studies indicating a probable better understanding of the disease management. The results could encourage physicians to use this facility when indicated for severely ill COVID-19 patients to save lives despite marked differences in practice and outcomes between different countries.

## Supporting information

**S1 Checklist. PRISMA 2009 checklist.**
(DOCX)

**S1 Table. Quality of the included studies; NOS = Newcastle Ottawa scale.**
(DOCX)

## Author Contributions

**Conceptualization:** Taha Aly Ahmed.

**Data curation:** Hany Hasan Elsayed, Aly Sherif Hassaballa, Taha Aly Ahmed, Hazem Youssef Sharkawy.

**Formal analysis:** Hany Hasan Elsayed, Aly Sherif Hassaballa, Mohammed Gumaa.

**Investigation:** Hazem Youssef Sharkawy.

**Methodology:** Taha Aly Ahmed, Mohammed Gumaa, Hazem Youssef Sharkawy.

**Supervision:** Hany Hasan Elsayed, Assem Adel Moharram.

**Writing – original draft:** Hany Hasan Elsayed, Aly Sherif Hassaballa, Mohammed Gumaa, Hazem Youssef Sharkawy.

**Writing – review & editing:** Hany Hasan Elsayed, Taha Aly Ahmed.

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
