## [Decision Letter · Decision Letter 0]

18 Feb 2021

PONE-D-20-28716

Variation in outcome of invasive mechanical ventilation between different countries for patients with severe COVID-19: a systematic review and meta-analysis

PLOS ONE

Dear Dr. Elsayed,

Thank you for submitting your manuscript to PLOS ONE. After careful consideration, we feel that it has merit but does not fully meet PLOS ONE’s publication criteria as it currently stands. Therefore, we invite you to submit a revised version of the manuscript that addresses the points raised during the review process.

The manuscript deals with quite a hot topic in everyday clinical practice. We suggest to discuss mortality rates (in respect to ventilatory strategy) of other disease states in respect to COVID disease. 

We look forward to receiving your revised manuscript.

Kind regards,

Chiara Lazzeri

Academic Editor

PLOS ONE

Journal Requirements:

2. Please attach a Supplemental file of the results a the quality assessment for each individual study assessed, reporting the outcome for each individual criteria considered.

3. We noticed you have some minor occurrence of overlapping text with the following previous publications, which needs to be addressed:

- https://associationofanaesthetists-publications.onlinelibrary.wiley.com/doi/full/10.1111/anae.15201

- https://www.atsjournals.org/doi/pdf/10.1164/rccm.202004-1385ED

- https://annalsofintensivecare.springeropen.com/articles/10.1186/s13613-020-00692-6

In your revision ensure you cite all your sources (including your own works), and quote or rephrase any duplicated text outside the methods section. Further consideration is dependent on these concerns being addressed.

Reviewers' comments:

Reviewer's Responses to Questions

**Comments to the Author**

1. Is the manuscript technically sound, and do the data support the conclusions?

Reviewer #1: Yes

2. Has the statistical analysis been performed appropriately and rigorously? 

Reviewer #1: Yes

3. Have the authors made all data underlying the findings in their manuscript fully available?

Reviewer #1: Yes

4. Is the manuscript presented in an intelligible fashion and written in standard English?

Reviewer #1: Yes

5. Review Comments to the Author

Reviewer #1: This is a very important topic for which a systematic review provides information that is of vital importance for clinical work and healthcare policy.

I would suggest editing the introduction so to make it current: it should definitely be mentioned that vaccination has begun as a vaccine has been approved and more will follow. A reference to an extremely recent meta-analysis titled Case Fatality Rates for Patients with COVID-19 Requiring Invasive Mechanical Ventilation. A Meta-analysis by Zheng Jie Lim et al. as well as discussion of its implications is warranted in the introduction section; if you wish to elaborate and contrast your findings with it, you could do so in the discussion.

It would be helpful to mention mortality rates after the initiation of mechanical ventilation for other conditions treated in the ICU other than the previous coronavirus diseases (eg. sepsis, pancreatitis, bacterial pneumonia, other viral pneumonias, COPD exacerbations) in the discussion. It is important to know whether the mortality of COVID19 when treated with mechanical ventilation is similar to other diseases which may be treated similarly in the ICU.

I would also suggest doing a sensitivity analysis by including the 6 studies which included only patients with a definitive outcome. Another sensitivity analysis that could be attempted is the exclusion of studies with a small number (eg. less than 100) of patients with a definitive outcome. This may lead to a result with lower heterogeinty.

6. PLOS authors have the option to publish the peer review history of their article (what does this mean?). If published, this will include your full peer review and any attached files.

Reviewer #1: No

---

## [Author Response · Author response to Decision Letter 0]

18 May 2021

Response to reviewers

Associate editor:

Thank you for praising our manuscript and providing precise and constructive recommendations for revision

Comment 1 Please attach a Supplemental file of the results a the quality assessment for each individual study assessed, reporting the outcome for each individual criteria considered.

Answer: A supplemental file is attached

Comment 2: We noticed you have some minor occurrence of overlapping text with the following previous publications, which needs to be addressed:

- https://associationofanaesthetists-publications.onlinelibrary.wiley.com/doi/full/10.1111/anae.15201

- https://www.atsjournals.org/doi/pdf/10.1164/rccm.202004-1385ED

- https://annalsofintensivecare.springeropen.com/articles/10.1186/s13613-020-00692-6

In your revision ensure you cite all your sources (including your own works), and quote or rephrase any duplicated text outside the methods section.

Changes: All mentioned sources have been addressed

Reviewer 1

Thank you for your valuable and constructive comments

Comment 1: I would suggest editing the introduction so to make it current: it should definitely be mentioned that vaccination has begun as a vaccine has been approved and more will follow.

Answer: The introduction has been edited and a paragraph about vaccination added.

Comment 2: A reference to an extremely recent meta-analysis titled Case Fatality Rates for Patients with COVID-19 Requiring Invasive Mechanical Ventilation. A Meta-analysis by Zheng Jie Lim et al. as well as discussion of its implications is warranted in the introduction section;

Answer 2: The reference has been added in the introduction and the implications of the study mentioned

Comment 3: It would be helpful to mention mortality rates after the initiation of mechanical ventilation for other conditions treated in the ICU other than the previous coronavirus diseases (eg. sepsis, pancreatitis, bacterial pneumonia, other viral pneumonias, COPD exacerbations) in the discussion. It is important to know whether the mortality of COVID19 when treated with mechanical ventilation is similar to other diseases which may be treated similarly in the ICU

Changes 3: A paragraph added in the discussion quoting mortality from other diseases treated in the ICU

Comment 4: I would also suggest doing a sensitivity analysis by including the 6 studies which included only patients with a definitive outcome. Another sensitivity analysis that could be attempted is the exclusion of studies with a small number (eg. less than 100) of patients with a definitive outcome. This may lead to a result with lower heterogeinty.

Changes 4: 2 new analysis have been added as required

---

## [Editor Report · Decision Letter 1]

24 May 2021

Variation in outcome of invasive mechanical ventilation between different countries for patients with severe COVID-19: a systematic review and meta-analysis

PONE-D-20-28716R1

Dear Dr. Elsayed,

We’re pleased to inform you that your manuscript has been judged scientifically suitable for publication and will be formally accepted for publication once it meets all outstanding technical requirements.

Kind regards,

Chiara Lazzeri

Academic Editor

PLOS ONE
---

## [Editor Report · Acceptance letter]

26 May 2021

PONE-D-20-28716R1 

Variation in outcome of invasive mechanical ventilation between different countries for patients with severe COVID-19: a systematic review and meta-analysis 

Dear Dr. Elsayed:

I'm pleased to inform you that your manuscript has been deemed suitable for publication in PLOS ONE. Congratulations! Your manuscript is now with our production department. 

Kind regards, 

on behalf of

Dr. Chiara Lazzeri 

Academic Editor

PLOS ONE